# Diversity and Seasonal Abundance of the Pine Bark and Ambrosia Beetles in the Florida Panhandle

**DOI:** 10.3390/insects16121275

**Published:** 2025-12-15

**Authors:** Ann Marie S. Robinson-Baker, Muhammad Haseeb, Lambert H. B. Kanga

**Affiliations:** Center for Biological Control, College of Agriculture and Food Sciences, Florida Agricultural and Mechanical University, Tallahassee, FL 32307, USA; annmarie1.robinson@famu.edu (A.M.S.R.-B.);

**Keywords:** ambrosia beetles, pine bark beetles, species diversity, seasonal abundance

## Abstract

Ambrosia and pine bark beetles pose significant threats to forest ecosystems in the Southeastern United States, contributing to tree decline, reduced timber quality, and widespread mortality. This study examined the seasonal activity of these beetles in Leon and Gadsden Counties, Florida. A total of 1657 individuals representing 18 genera and 24 species were captured using baited Lindgren funnel traps. Although hand sanitizer functions as a broad-spectrum attractant, traditional semiochemical lures such as α-pinene, a monoterpene naturally emitted from pine resin, and species-specific pheromones like ipsdienol and frontalin provide more targeted attraction for monitoring specific taxa, including *Ips* species and the southern pine beetle. However, these specialized lures can be costly and often require controlled-release dispensers. To establish a more economical and practical monitoring system, this study utilized hand sanitizer as a lure to assess beetle abundance and diversity across both counties. This approach proved efficient, easy to deploy, and effective for capturing a wide range of ambrosia and pine bark beetles. The findings demonstrate that hand sanitizer-baited traps offer a valuable, cost-effective tool for surveying species distribution and tracking population dynamics, particularly in regions where budget limitations may restrict the use of more specialized attractants.

## 1. Introduction

The timber industry contributes approximately 4% of the total United States (U.S.) manufacturing Gross Domestic Product (GDP), generating nearly $300 billion in products and services annually [1]. Florida encompasses 17.16 million acres (68,795 square km) of forested land, accounting for 50% of the state’s total land area [2]. Forest products support the production of over 5000 items that benefit humans [3,4].

Pine bark beetles (PBBs) are among the most significant biotic agents influencing forest dynamics, annually infesting vast forest areas and altering community composition and structure. The seasonal abundance of pine bark beetles in Florida aligns with climatic seasonality, with beetle activity peaking during warmer months. Typically, the highest beetle populations occur from spring to summer (April to August), coinciding with higher temperatures and drought conditions that stress pine trees. These environmental factors trigger the release of host volatiles, such as ethanol and α-pinene, making this period the most critical for monitoring efforts. Pest outbreaks may increase harvesting costs, accelerate cutting schedules, and reduce timber value, revenue, and yield [5,6]. These impacts highlight the extensive ecological, economic, and social consequences of bark beetle infestations on forest ecosystems [7,8]. Among these beetles, the southern pine beetle (*Dendroctonus frontalis* Zimmermann, 1868) is considered one of the most destructive pests of pine forests in the Southeastern U.S. [9,10,11]. Historically, aerial and ground surveys have documented severe damage caused by PBB outbreaks. Between 1999 and 2002, an outbreak in the eastern U.S. caused timber losses exceeding one billion dollars, while over 91,054 hectares of pine forests in Central America were also extensively damaged during the same period [9]. In the western U.S., since 2000, outbreaks of various PBBs have affected over 36 million hectares of forest [9,10,11,12,13,14]. Climate strongly influences species distribution, abundance, and potential range by affecting survival, growth, and reproduction [15,16]. The activity and abundance of PBBs are strongly affected by environmental factors such as temperature, solar radiation, and humidity. Fluctuations in these variables can alter beetle life cycles, reproduction rates, and survival. As climate conditions shift, driven by extreme events such as hurricanes and by human-induced ecological changes, abiotic factors may increasingly favor PBB proliferation [17,18]. This can lead to high infestations and greater damage to pine forests, which are highly susceptible to attack.

PBB populations often fluctuate dramatically and can rapidly emerge as major pests in coniferous forests [19,20]. Indeed, their dynamics are primarily influenced by host tree susceptibility, a process driven by bottom-up ecological forces [21,22,23]. Monitoring typically involves the use of synthetic aggregation pheromones in combination with Lindgren funnel traps, which attract beetles and allow researchers to assess population trends and confirm infestation levels [24,25]. While effective, this method is often complemented by aerial surveys using drones, satellites, or manned aircraft equipped with multispectral or infrared imaging to detect large-scale tree mortality. However, aerial surveys are less precise in identifying early-stage infestations, underscoring the need for integrated monitoring approaches [26].

Ambrosia beetles maintain a unique symbiotic relationship with fungi, which they cultivate in their galleries as a food source for adults and larvae. This mutualism allows them to colonize a wide range of host trees and thrive even in nutrient-poor wood [27]. The fungi are typically transported in specialized structures called mycangia, and infestations can lead to xylem blockage, tree decline, and eventually tree mortality [28]. Some species, such as *Xyleborus glabratus*, *Xylosandrus crassiusculus*, and *Cnestus mutilatus*, are recognized invasive pests in the southeastern United States, where they have been associated with widespread mortality of redbay, avocado, and other hardwoods through the transmission of pathogenic fungi like *Raffaelea lauricola*, the causal agent of laurel wilt disease [29,30].

In recent years, human activities have accelerated changes in landscape structure, species distributions, and climate, with far-reaching implications for biodiversity, ecosystem functioning, and forest resilience [20,31,32]. To study the seasonal dynamics of pests such as and ambrosia and PBBs, traps are typically deployed from early spring through summer [33,34,35]. During unfavorable winter conditions, PBBs enter diapause [34,36]. This adaptation enables survival until favorable conditions return in early spring, when flight activity resumes [37,38,39,40]. Throughout these phases, environmental cues regulate the beetles’ life cycle, influencing population fluctuations and the extent of forest damage [41,42]. Although temperature is a key driver of these patterns, their natural enemies play only a minor role in regulating PBB populations [43].

This study was conducted to quantify ambrosia beetles and PBBs in pines in the Leon and Gadsden Counties of the Florida Panhandle. We utilized ethanol-based hand sanitizer as a lure, a practical and cost-effective alternative to synthetic semiochemicals. Also, this study evaluated the species composition and determined the relationships between beetle abundance and climatic variables using cost-effective monitoring tools.

## 2. Materials and Methods

### 2.1. Study Area

This experiment was carried out in the Florida Panhandle in two distinct locations: the Florida A&M University (FAMU) Research and Extension Center in Quincy (30.4247° N, 84.2846° W) and the Center for Viticulture and Small Fruit Research (30.4767° N, 84.1726° W). The pine stand in Quincy consists of a well-established and maintained 4-acre stand of Loblolly pine (*Pinus taeda* L.). The ages of the trees were over 23 years (Figure 1). Similarly, the pine stand at the Center for Viticulture and Small Fruit Research was also composed of Loblolly pine (*Pinus taeda* L.) of the same age, providing a complementary study environment for assessing pine bark beetle diversity and seasonal abundance under different ecological conditions.

A preliminary site evaluation was carried out at each study site to select suitable forest areas for trapping. Both sites were situated adjacent to major highways used by log trucks. Freshly cut timber was transported along these roads from state to state, playing a significant role in the widespread dissemination of the PBBs.

The Leon County site is characterized by relatively denser pine stands with a higher abundance of loblolly and slash pine, which provide ample host materials and resin-rich environments favorable to ambrosia beetles such as *Xyleborinus saxesenii*, *Cnestus mutilatus*, and *Xylosandrus crassiusculus*. In contrast, the Gadsden County sites appear to have more heterogeneous forest conditions, including mixed hardwood pine stands and slightly more fragmented habitats. Such differences may favor other beetle taxa, such as *Xylosandrus amputates*.

Climatic and topographic variations may also influence beetle growth and development. The Leon County sites is generally at lower elevation with more humid microclimates, while Gadsden County includes slightly higher, hillier terrain that may influence host stress levels and beetle flight dynamics. These subtle ecological variations, combined with differences in host tree density and age class, likely drive the observed contrast in species composition and dominance. The bark and ambrosia beetles feed and reproduce within the bark and wood of their hosts, which buffers them against short-term weather fluctuations. As a result, local forest structure and host tree stress often exert a stronger influence on community patterns than ambient climate conditions.

### 2.2. Monitoring, Collection, and Identification of the Pine Bark Beetles in Two Counties

Six commercial Lindgren funnel traps (1.2 m above the ground) were set in pine stands in Leon and Gadsden counties, with three traps per site. All traps were baited with Purell^®^ Advanced Hand Sanitizer (70% ethanol), since ethanol mimics stress volatiles that attract bark beetles [44,45,46,47]. The study ran in the Florida Panhandle from 8 July 2022 to 25 October 2023, with collections occurring twice weekly and monthly records of species were recorded. Beetles captured in the traps were removed using a soft spatula to prevent damage. These beetles were transferred to mason jars with labels containing 95% ethanol. Then, the beetles were transported to the CBC laboratory at Florida A&M University and examined under a Leica MZ16 stereomicroscope (Leica, Wetzlar, Germany). The specimens were preserved in 98% ethanol. The specimens were mounted on pins for imaging. In the lab, beetle identifications were carried out based on A Guide to Florida’s Common Bark and Ambrosia Beetles [48]. Final confirmation of species was determined by Dr. Paul Skelley at FDACS-DPI, Gainesville [44,45,46,47,48].

### 2.3. Seasonal Data

The meteorological data were collected from the publicly accessible Weather Underground (WU) database. For accuracy and relevance, the data sets were obtained from the weather stations closest to the beetle sampling sites. These data points were then subjected to statistical tests to assess the relationship between PPB population dynamics and environmental conditions, providing insights into how climatic variables influence beetle behavior and population trends.

For the study site located in Leon County, data were collected from the Center for Viticulture and Small Fruit Research. The environmental data for the site were sourced from the Florida State University’s WeatherSTEM station (station ID: KFLTALLA154), situated at 30.44° N, 84.30° W, and an elevation of 8.5 m.

For the research site located in Gadsden County at the FAMU Research and Extension Center in Quincy (30.4247° N, 84.2846° W), meteorological data were obtained from the High Bridge Rd weather station (station ID: KFLQUINC12), situated at an elevation of 78 m. By aligning the sampling sites with these nearby weather stations, we ensured that the environmental data sets collected accurately represented the conditions at each location. This allowed for a more precise analysis of ecological factors influencing PBB population dynamics across various study sites in the Florida Panhandle.

### 2.4. Data Analysis

The density and seasonal abundance of ambrosia and PBBs were analyzed in relation to key environmental variables; temperature, relative humidity, and precipitation during 2022 and 2023. Data were processed and statistically evaluated using Generalized Linear Models (GLM) and Excel, focusing on the strength and direction of associations between beetle population dynamics and climatic conditions.

To explore multivariate relationships between beetle abundance and environmental factors, a Principal Component Analysis (PCA) was conducted. Insect abundance data were standardized (mean = 0, standard deviation = 1) prior to analysis to minimize the effects of scale differences among variables. The PCA incorporated three climatic variables: relative humidity (%), temperature (°C), and precipitation (mm) as environmental gradients influencing species distribution. Variable loadings were visualized as vectors (arrows) in the ordination biplot, indicating the direction and magnitude of their correlation with beetle assemblages.

All statistical analyses were performed using SAS version 9.4 (SAS Institute Inc., Cary, NC, USA), employing the PROC PRINCOMP procedure to generate principal components and estimate correlations between environmental variables and insect abundance.

## 3. Results

### 3.1. Species Diversity and Abundance

Between 8 July 2022 and 25 October 2023, a total of 1594 pine bark and ambrosia beetles were captured in Leon County, representing 24 species and 18 genera across eight tribes: Xyleborini, Corthylini, Cryphalini, Hylurgini, Ipini, Onycholipini, Platypodinae, and Ploeotribini. The taxonomic placement of most captured beetles followed the classification system outlined by Haack & Rabaglia [49] The frequency distribution of these species is summarized in Table 1 and illustrated in Figure 2, showing a wide variation in species abundance ranging from 0.06% to 28.96%.

The dominant species was *Xyleborinus saxesenii*, which accounted for 28.96% of all individuals captured, confirming its strong presence in the region. Two subdominant species, *Cnestus mutilatus* (15.24%) and *Xylosandrus crassiusculus* (14.27%), also contributed substantially to the total number of beetles caught. In contrast, several species such as *Ambrosiophilus atratus*, *Hypothenemus interstitialis*, *Ips grandicollis*, *Xylosandrus amputatus*, and *Cryptocarenus heveae* occurred in much lower relative abundances, each contributing between 0.06% and 9% of the total beetles caught. These results indicate that the community structure of PBBs in Leon County is skewed toward a few dominant ambrosia beetle species, with the remaining species representing a less common but ecologically diverse assemblage.

In addition to bark beetles, the traps occasionally collected other wood- and bark-associated insects, including *Pseudopentarthrum atrolucens*, *Metachroma* sp., *Euvrilletta* sp., *Xylobiops basilaris*, *Cossonus impressus*, and *Lobogestoria gibbicollis*. Notably, two species, *Cossonus impressus* and *Lobogestoria gibbicollis*, represent new county records for Leon County (Table 2). This highlights the value of long-term trapping for not only monitoring PBB populations but also detecting other rare or previously unrecorded forest insects.

During the survey conducted in Gadsden County, 63 pine bark and ambrosia beetles were captured from 16 May 2023 to 4 October 2023. These beetles represented 12 species within 10 genera, and their ranking based on their dominance is provided in Table 3. Throughout this collection period, the dominant species observed was *Xylosandrus amputatus,* representing 25.4% of the beetles captured. Additionally, three subdominant species were identified: *Xyleborinus saxesenii* accounted for 20.63%, *Ambrosiodmus lewisi* 12.69%, and *Xylosandrus compactus* 11.11% of the captured beetles (Table 3, Figure 3). These species represent key players in the beetle populations within Gadsden County. Their relative abundances suggested significant ecological roles in the local environment, potentially affecting tree health and forest dynamics.

The analysis using Generalized Linear Models (GLMs) showed that insect abundance was significantly associated with the climatic variables evaluated. The Poisson distribution model provided the best fit (Pseudo *R*^2^ = 1.000), adequately explaining the observed variation in the beetle counts.

Among the climatic factors, both relative humidity and temperature had significantly negative effects on abundance. For each 1% increase in humidity, the predicted insect abundance decreased by approximately 5.6%, while a 1 °C increase in temperature resulted in an estimated 7.1% reduction in the beetle numbers. Conversely, precipitation had a significantly positive effect: each 1 mm increase in rainfall promoted an increase of about 0.22% in the beetle abundance (Figure 4).

In addition, the marginal effect plots confirmed these trends. The beetle abundance was highest at low temperature and low humidity, with a gradual increase associated with the higher level of precipitation. The three-dimensional plot combining temperature and humidity showed that the lowest abundance occurs in warmer and more humid environments, while the highest beetle abundance was observed under the cooler and drier conditions.

The analysis of the Gadson County dataset using a GLM with Poisson distribution showed distinct patterns due to the effects of climatic variables on the beetle abundance. Relative humidity had a positive effect, with an exponential increase in predicted abundance as humidity rose, indicating that small increments in humidity resulted in substantial growth in the beetle density. Temperature also showed a positive trend, with gradual increases in abundance as temperature increased. However, this effect was less pronounced and not statistically significant. In contrast, precipitation had a marked negative effect, with a sharp decline in predicted beetle abundance as rainfall increased, suggesting that wetter environments reduced beetle abundance. Overall, the results indicated that beetle abundance was favored by warmer and more humid conditions but decline under higher rainfall levels (Figure 5).

In Leon County, the PCA revealed that the moisture-related variables relative humidity and precipitation were the primary drivers of beetle assemblage structure, while temperature contributed a secondary, opposing gradient. *Xyleborinus saxesenii* and *Xylosandrus crassiusculus* were strongly associated with high humidity and rainfall, indicating a preference for wetter conditions. *Cryptocarenus heveae* showed a weaker but still positive association with moisture. In contrast, *Cnestus mutilatus* aligned most closely with the temperature vector, reflecting its tendency to be more abundant under relatively warmer and drier conditions. *Hypothenemus interstitialis* and the mixed-species “Other” group occupied positions near the ordination center, suggesting broad ecological tolerances or weak climatic associations. Overall, moisture emerged as the dominant factor shaping species composition, with temperature exerting a secondary influence (Figure 6).

In Gadsden County, the PCA indicated that moisture, particularly relative humidity and closely followed by precipitation, was the dominant environmental factor affecting beetle assemblages. *Xyleborinus saxesenii* showed the strongest association with high humidity and rainfall, reflecting a clear preference for wetter conditions. *Xylosandrus amputatus* and *Xyleborus bispinatus* displayed moderate alignment with this same moisture gradient, while *Xylosandrus compactus* also occupied the wet-climate quadrant but with weaker influence due to its lower abundance. *Hypothenemus interstitialis* plotted near the ordination center, indicating broad tolerance or limited climatic sensitivity. The mixed “Other” category showed only modest association with moisture, consistent with its heterogeneous species composition. Overall, the Gadsden assemblage suggests that humid, rainy periods disproportionately favor *X. saxesenii* and, to a lesser extent, *X. amputatus*, *X. bispinatus*, and *X. compactus*, highlighting moisture as the key driver of species activity in this county (Figure 7).

The dorsal and lateral habitus of the beetle species collected and identified are provided in Figure 8.

### 3.2. Seasonal Fluctuation of Beetles in the Florida Panhandle

The relationship between the number of beetles caught and various environmental factors over time from July 2022 to October 2023 is provided in Figure 9. The background bar chart represents environmental factors, including temperature, humidity, and precipitation. The light blue bars indicate temperature in Celsius (°C), which ranged between approximately 13 °C and 29 °C throughout the months. The brown bars represent precipitation in millimeters, showing relatively low values across the months. The dark blue bars represent humidity as a percentage, which ranged from 64.42% to 83.6%.

The beetle density showed high activity for most species, particularly *Xyleborinus saxesenii* and *Cnestus mutilates*, under the moderate temperature ranges in the spring. This pattern suggested an optimal spring temperature range for beetle activity in Leon County, with populations declining as temperatures rise in the summer. Additionally, the high humidity levels in the spring appeared to have positively influenced beetle activity, especially for *Xyleborinus saxesenii* and *Xylosandrus crassiusculus*, which may have a preference or dependency on higher moisture levels.

## 4. Discussion

This study provides new insights into the diversity, seasonal dynamics, and climatic responses of pine bark and ambrosia beetles in the Florida Panhandle. Twenty-four species from 18 genera were recorded, with strong dominance by the tribe Xyleborini, consistent with earlier findings on this group’s ecological versatility and ability to exploit diverse woody substrates under changing environmental conditions [27,28,50]. Species abundance differed sharply between counties: Leon County yielded 1594 specimens versus only 63 in Gadsden County, despite identical sampling efforts. This disparity likely reflects differences in forest structure, microclimate, host availability, or recent disturbance events.

Hand sanitizer, due to its ethanol content, proved to be an effective, low-cost, and broadly attractive lure for monitoring bark beetles, mimicking stress volatiles emitted by weakened trees [35]. While traditional semiochemical lures α-pinene, ipsdienol, and frontalin provide more targeted monitoring for taxa such as Ips and southern pine beetles [11], they are more expensive and require specialized dispensers. Hand sanitizer therefore offers a practical alternative for general surveillance, with enhanced detection possible by combining ethanol with α-pinene or species-specific pheromones.

The results from Leon County and Gadsden County highlight contrasting relationships between insect abundance and climatic variables, indicating that insect responses to environmental factors can vary considerably across spatial and temporal scales. In Leon County, both relative humidity and temperature had significant negative effects, while precipitation had a positive effect, suggesting that insect populations were more abundant under cooler, drier conditions with moderate rainfall. This pattern is consistent with studies showing that high temperatures and excessive humidity can reduce insect survival and activity by increasing physiological stress and enhancing the risk of fungal pathogens [51,52]. Precipitation, in turn, may promote vegetation growth and resource availability, indirectly supporting insect populations [53,54,55,56,57,58].

Community composition showed clear dominance patterns. *Xyleborinus saxesenii* was the most abundant species overall, particularly in Leon County, reflecting its generalist habits and strong colonization potential in subtropical pine systems [27,28]. In contrast, *Xylosandrus amputatus* was most prevalent in Gadsden County, likely influenced by localized host associations or environmental conditions [24]. Other common species *Cnestus mutilatus*, *Xylosandrus crassiusculus*, and *Ambrosiodmus lewisi* underscore the presence of multiple invasive or economically significant taxa. Low-abundance species such as *Ips grandicollis* and *Euplatypus compositus* contributed to overall richness and highlight the value of sustained monitoring [42].

Nevertheless, longer-term drought stress remains a well-known contributor to host susceptibility, as weakened trees produce fewer defensive resins. The detection of *Ambrosiodmus lewisi* and *Cyclorhipidion distinguendum* as new county records highlights ongoing expansions potentially driven by climate change and timber transport networks [27,28]. These findings reinforce the need for long-term, spatially extensive monitoring to refine risk models and track climatic influences on scolytine populations [47].

Multivariate analyses (PCA and GLM) showed that moisture-related variables particularly relative humidity and precipitation were the dominant environmental gradients shaping assemblage patterns. In Leon County, humidity and precipitation were strongly aligned with species such as *X. saxesenii* and *X. crassiusculus*, consistent with reports that fungal symbiont viability and gallery success improve under humid conditions [50,59]. *Cnestus mutilatus* associated more closely with temperature, reflecting its adaptation to warmer and drier settings [24]. Species such as *Cryptocarenus heveae* and *Hypothenemus interstitialis* exhibited broader ecological tolerances, aligning with their polyphagous habits.

In Gadsden County, humidity emerged as the most influential variable, with *X. saxesenii*, *X. amputatus*, *X. compactus*, and *X. bispinatus* showing positive associations with wetter conditions [59,60,61,62]. Differences between counties likely reflect variation in forest structure, microclimate, and host composition, demonstrating that beetle–climate interactions are spatially context-dependent [52,57]. Microhabitat conditions such as canopy density, soil moisture, and vegetation diversity can moderate climatic effects, influencing both host tree vulnerability and beetle colonization success [56,63].

Overall, moisture availability and host tree condition appear to be the primary determinants of bark and ambrosia beetle community dynamics in the region. While temperature influences physiology and dispersal, humidity and rainfall regulate fungal symbioses essential for ambrosia beetle success. As climate change alters humidity and rainfall patterns, long-term monitoring will be essential for improving outbreak prediction models and guiding adaptive forest management strategies in Florida’s pine ecosystems [58,64].

## 5. Conclusions

This study documented the diversity and seasonal abundance of ambrosia and pine bark beetles in the Florida Panhandle, capturing 1657 specimens representing 24 species and 18 genera. Three new county records were identified *Xylosandrus amputatus* and *Ambrosiodmus lewisi* in Leon County and *Cyclorhipidion distinguendum* in Gadsden County highlighting ongoing shifts in species distributions. Beetle activity peaked during warmer months, reflecting their dependence on favorable climatic conditions and host availability.

The detection of new county records further illustrates the influence of trade, human movement, and environmental change on species introductions. Strengthening surveillance by combining cost-effective ethanol-based lures with species-specific pheromones will improve monitoring precision. As climate change continues to alter forest conditions, refining detection, monitoring, and management strategies will be essential to mitigate future outbreaks, protect biodiversity, and reduce economic losses. Integrating affordable monitoring tools with ecological modeling will enhance predictive capacity and support proactive, climate-resilient forest management across Florida’s pine ecosystems.

## Figures and Tables

**Figure 1 insects-16-01275-f001:**
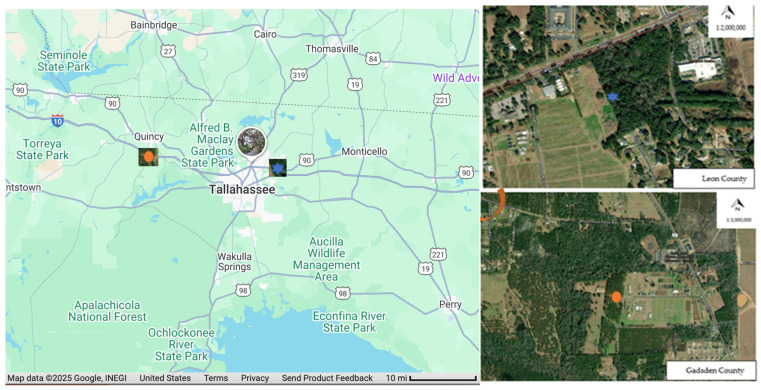
Map of the study sites with a blue star showing the location of Leon County site and orange oval showing the Gadsden County site. Source: Google Maps.

**Figure 2 insects-16-01275-f002:**
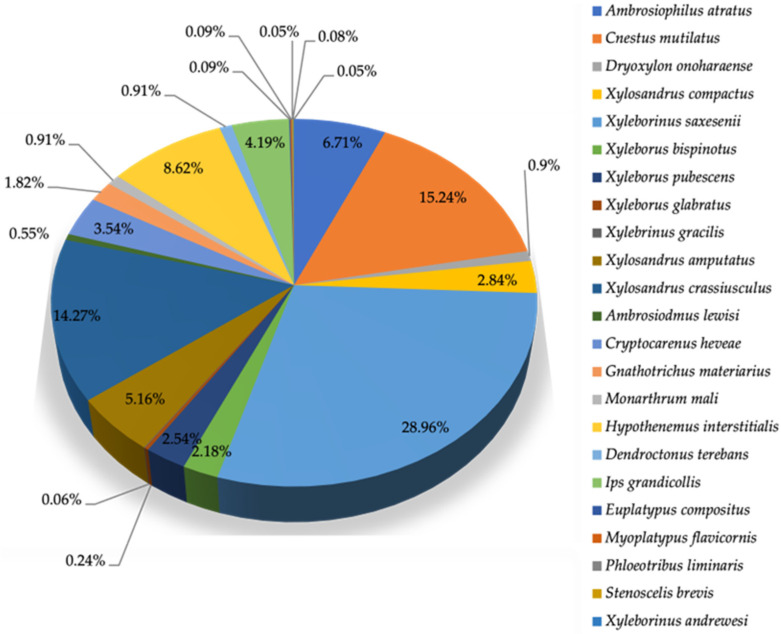
Relative abundance (%) of the dominant and sub-dominant species of pine bark and ambrosia beetles captured in Leon County (N = 1594).

**Figure 3 insects-16-01275-f003:**
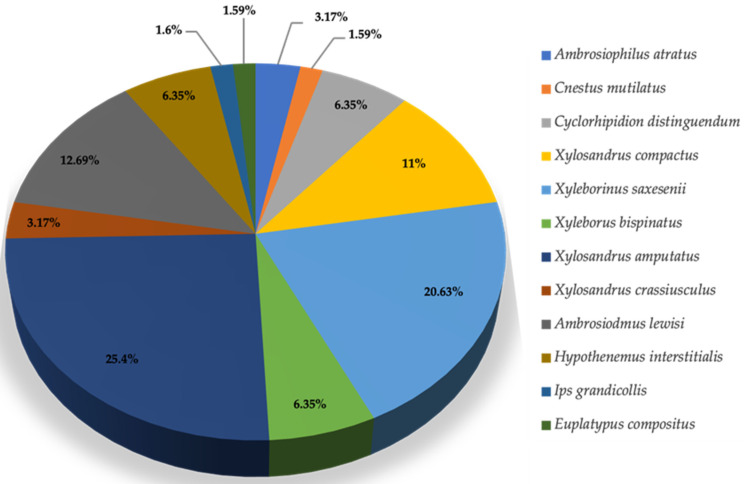
Relative abundance (%) representation for dominant and sub-dominant species of pine bark and ambrosia beetles collected from Gadsden County (N = 63).

**Figure 4 insects-16-01275-f004:**
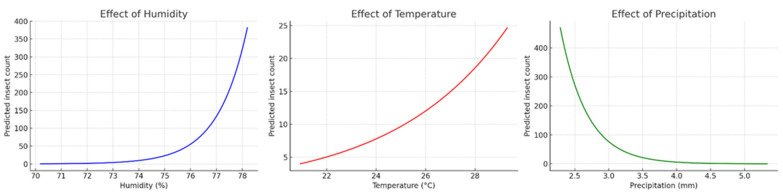
Marginal effects of climatic variables on insect abundance in Leon County based on a Poisson Generalized Linear Model (GLM). Predicted insect counts decreased with increasing relative humidity (**left**) and temperature (**center**), while precipitation exerted a positive effect, with insect abundance gradually increasing as rainfall increased (**right**).

**Figure 5 insects-16-01275-f005:**
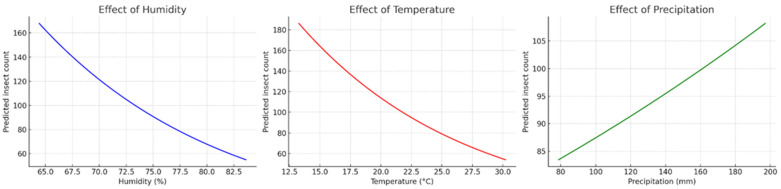
Marginal effects of climatic variables on the beetle abundance in Gadsden County based on a Poisson Generalized Linear Model (GLM). Predicted insect counts increased exponentially with higher relative humidity (**left**) and showed a positive and moderate trend with the changing temperature (**center**). In contrast, precipitation has a strong negative effect, with the beetle abundance decreasing sharply as rainfall increased (**right**).

**Figure 6 insects-16-01275-f006:**
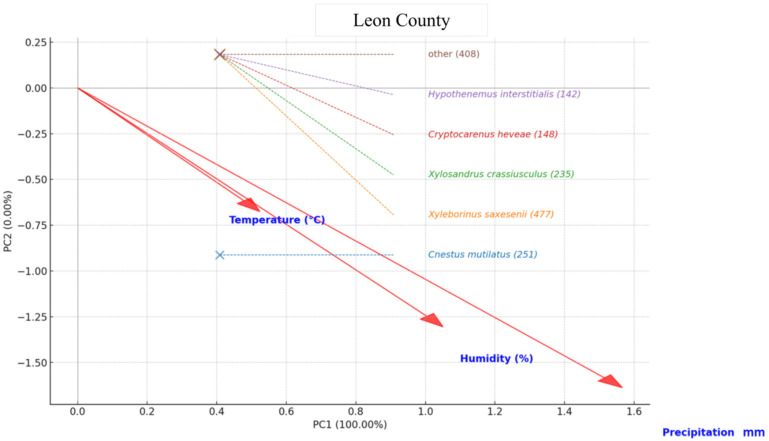
Principal Component Analysis (PCA) biplot showing the relationship between pine bark beetle species abundance and climatic variables (Temperature (°C), Humidity (%), and Precipitation (mm)) in Leon County. The species names are presented with their total abundance in parentheses, while red arrows indicate the direction and magnitude of climatic variables’ influence on species distribution.

**Figure 7 insects-16-01275-f007:**
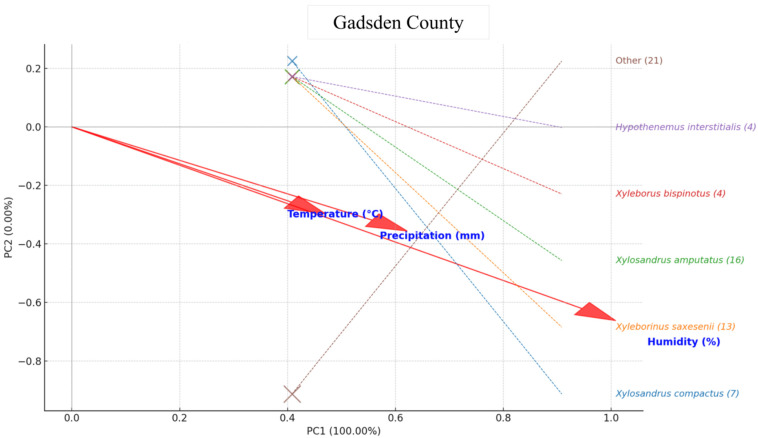
PCA biplot for Gadsden County: points = species (size ∝ abundance) with labels on the right; red arrows = climate variables (Temperature (°C), Humidity (%), Precipitation (mm)). Arrow direction indicates species association, arrow length indicates strength, and the angle between arrows indicates correlation. Axes = PC1 and PC2 based on standardized abundances.

**Figure 8 insects-16-01275-f008:**
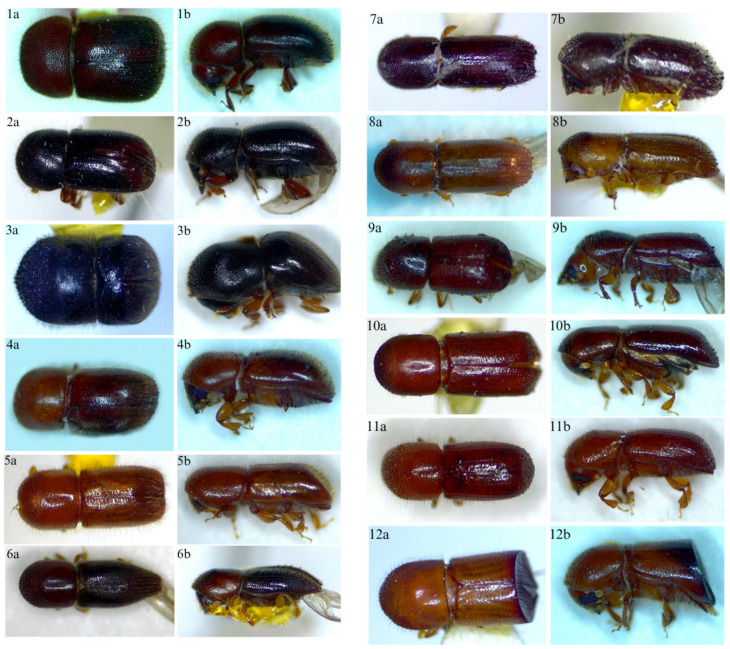
Dorsal and lateral habitus of collected and identified beetles (*A. lewisi* 1a,1b), (*A. astratus* 2a,2b), (*C. mutilatus* 3a,3b), (*C. distinguendum* 4a,4b), *D. onoharaense* 5a,5b), (*X. andrewesi* 6a,6b), (*X. gracillis* 7a,7b), (*X. saxesenii* 8a,8b), (*X. bispinatus* 9a,9b), (*X. glabratus* 10a,10b), (*X. pubescens* 11a,11b), (*X. amputatus* 12a,12b), (*C. heaveae* 13a,13b), (*G. materiarius* 14a,14b), (*M. mali* 15a,15b), (*H. interstitialis* 16a,16b), (*D. terebrans* 17a,17b), (*I. grandicollis* 18a,18b), (*S. brevis* 19a,19b), (*E. compositus* 20a,20b), (*M. flavicornis* 21a,21b), (*P. liminaris* 22a,22b), (*X. crassiusculus* 23a,23b), (*X. compactus* 24a,24b).

**Figure 9 insects-16-01275-f009:**
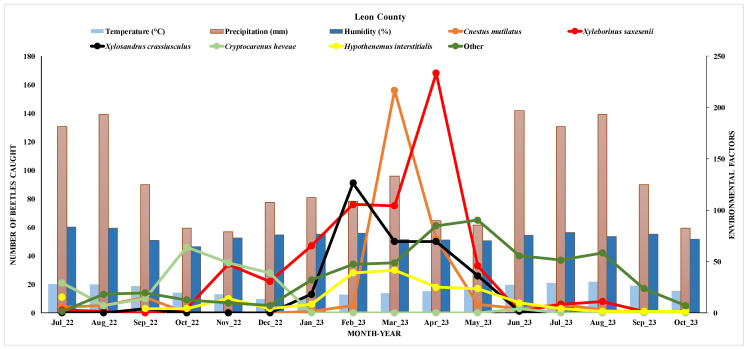
Seasonal distribution of the dominant species of the ambrosia and pine bark beetles and effects of environmental factors in Leon County.

**Table 1 insects-16-01275-t001:** Relative abundance of the pine bark and ambrosia beetle species collected in Leon County from 8 July 2022 to October 2023 (N = 1594).

Tribe	Scientific Name	Common Name	Relative Abundance	Rank
Xyleborini				
	*Ambrosiodmus lewisi* (Blandford, 1894)	Ambrosia beetle	0.55%	Subrecedents
	*Ambrosiophilus atratus* (Eichhoff, 1876)		6.71%	Dominants
	*Cnestus mutilatus* (Blandford, 1894)	Camphor shoot borers	15.24%	Dominance
	*Dryoxylon onoharaense* (Eggers, 1930)		0.90%	Subrecedents
	*Xyleborinus andrewesi* (Blandford, 1896)		0.05%	Subrecedents
	*Xyleborinus gracilis* (Eichhoff 1868)		0.06%	Subrecedents
	*Xyleborinus saxesenii* (Ratzeburg, 1837)	Fruit-tree pinhole borer	28.96%	Dominance
	*Xyleborus bispinatus* (Eichhoff, 1868)		2.18%	Subdominants
	*Xyleborus glabratus* (Eichhoff, 1868)	Redbay ambrosia beetle	0.24%	Subrecedents
	*Xyleborus pubescens* (Zimmermann, 1868)		2.54%	Subdominants
	*Xylosandrus amputatus* (Blandford, 1894)	Asian ambrosia beetle	5.16%	Dominants
	*Xylosandrus compactus* (Eichhoff 1876)	Black twig borer	2.84%	Subdominants
	*Xylosandrus crassiusculus* (Motschulsky, 1866)	Granulate ambrosia beetle	14.27%	Dominance
Corthylini	*Cryptocarenus heveae* (Hagedorn, 1912)		3.54%	Subdominants
	*Gnathotrichus materiarius* (Fitch, 1858)	Pine timber-beetle or timber-beetle	1.82%	Recedants
	*Monarthrum mali* (Fitch, 1858)	Apple wood Stainer	0.91%	Subrecedents
Cryphalini	*Hypothenemus interstitialis* (Hopkins, 1915)		8.62%	Dominants
Hylurgini	*Dendroctonus terebans* (Olivier, 1795)	Black turpentine beetle	0.91%	Subrecedents
Ipini	*Ips grandicollis* (Eichhoff, 1868)		4.19%	Subdominants
Onycholipini	*Stenoscelis brevis* (Boheman, 1845)		0.08%	Subrecedents
Platypodini	*Euplatypus compositus* (Say, 1823)		0.09%	Subrecedents
	*Myoplatypus flavicornis* (Fabricius, 1776)		0.09%	Subrecedents
Phloeotribini	*Phloeotribus liminaris* (Harris, 1852)	Peach bark beetle	0.05%	Subrecedents

**Table 2 insects-16-01275-t002:** Other species of beetles captured in traps in the Leon and Gadsden Counties during this study.

Family Name	Tribe	Scientific Name	Common Name	Leon	Gadsden
Curculionidae	Onycholipini	*Pseudopentarthrum atrolucens*		7	
Curculionidae	Cossonini	*Cossonus impressus*		27	2
Chrysomelidae	Typophorini	*Metachroma* sp.	Leaf beetle	1	1
Zopheridae	Synchitini	*Lobogestoria gibbicollis*	Cylindrical bark beetle	1	-
Ptinidae	Xyletinini	*Euvrilletta* sp.	Anobiid powderpost beetle	2	-
Bostrichidae	Xyloperthini	*Xylobiops basilaris*	Red-shouldered shothole borer	5	-
Staphylinidae	Oxypodini	*Nicrophorus sayi*	Carrion Beetle		-

**Table 3 insects-16-01275-t003:** Relative abundance of the pine bark and ambrosia beetle species collected in Gadsden County from 16 May 2023 to 4 October 2023 (N = 63).

Tribe	Scientific Name	Common Name	Relative Abundance	Rank
Xyleborini	*Ambrosiophilus atratus* (Eichhoff, 1876)		3.17%	Subdominants
	*Cnestus mutilatus* (Blandford, 1894)	Camphor shoot borers	1.59%	Recedants
	*Cyclorhipidion distinguendum* (Eggers, 1930)		6.35%	Dominants
	*Xylosandrus compactus* (Eichhoff, 1876)	Black twig borer	11.11%	Dominance
	*Xyleborinus saxesenii* (Ratzeburg 1837)	Fruit-tree pinhole borer	20.63%	Dominance
	*Xyleborus bispinatus* (Eichhoff, 1868)		6.35%	Dominants
	*Xylosandrus amputatus* (Blandford, 1894)	Asian ambrosia beetle	25.40%	Dominance
	*Xylosandrus crassiusculus* (Motschulsky, 1866)	Granulate ambrosia beetle	3.17%	Subdominants
	*Ambrosiodmus lewisi* (Blandford, 1894)	Ambrosia beetle	12.69%	Dominance
Cryphalini	*Hypothenemus interstitialis* (Hopkins, 1915)		6.35%	Dominants
Ipini	*Ips grandicollis* (Eichhoff, 1868)		1.60%	Recedants
Platypodini	*Euplatypus compositus* (Say, 1823)		1.59%	Recedants

## Data Availability

The data presented in this study are available on request from the corresponding author.

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
