# Peer review of "Diversity and Seasonal Abundance of the Pine Bark and Ambrosia Beetles in the Florida Panhandle"

_insects, 2025, doi:10.3390/insects16121275_

Round 1

Reviewer 1 Report (Previous Reviewer 2)

Comments and Suggestions for Authors

Thank you for your hard work revising the manuscript. It has greatly improved the document.

Unfortunately, the abstract primarily focuses on the use of attractants to collect the beetles and does not highlight the many new and interesting elements developed in the revised version of the manuscript. A few additional words would be welcome.

The conclusions are consistent with the evidence and arguments presented, and address the central question. Climate analysis revealed that humidity-related variables, particularly relative humidity and precipitation, played a major role in the formation of beetle assemblages, while the effects of temperature varied between sites.

The references are appropriate (with a few necessary corrections to the style; see additional comments below) and I have no further comments on the tables and figures.

Other comments:

Standardise the references to journal titles by following the instructions for authors (note that some titles are abbreviated and some are not).

Line 762 (reference 19). Write the authors' names in lowercase.

Line 901 (reference 59): Write the sentence in lowercase.

Author Response

Thank you for your hard work revising the manuscript. It has greatly improved the document.

Unfortunately, the abstract primarily focuses on the use of attractants to collect the beetles and does not highlight the many new and interesting elements developed in the revised version of the manuscript. A few additional words would be welcome.

Response: Thank you for your time and review of the manuscript. We greatly appreciate this. As advised, we have updated the abstract with additional information.

The conclusions are consistent with the evidence and arguments presented, and address the central question. Climate analysis revealed that humidity-related variables, particularly relative humidity and precipitation, played a major role in the formation of beetle assemblages, while the effects of temperature varied between sites.

The references are appropriate (with a few necessary corrections to the style; see additional comments below) and I have no further comments on the tables and figures.

Other comments:

Standardise the references to journal titles by following the instructions for authors (note that some titles are abbreviated and some are not).

Response: The references section has been revisited and revised. 

Line 762 (reference 19). Write the authors' names in lowercase.

Response: Thank you. This has been modified to a lowercase.

Line 901 (reference 59): Write the sentence in lowercase.

Response: Thank you. This has been modified to a lowercase

Reviewer 2 Report (Previous Reviewer 3)

Comments and Suggestions for Authors

This manuscript investigates the diversity and seasonal abundance of pine bark and ambrosia beetles in the Florida Panhandle, using ethanol-based hand sanitizer as a lure. The topic is relevant to forest entomology and pest management, and the dataset provides valuable regional information. The approach is original and potentially useful for cost-effective monitoring of Scolytinae species.

While the study addresses an important subject, the manuscript requires major revision to improve clarity, organization, and consistency. Several sections repeat similar information, and the overall structure would benefit from careful reorganization to enhance flow and eliminate redundant explanations. Figures, tables, and statistical interpretations should be reviewed and refined to ensure that the results are presented clearly and accurately.

Figure 2 could be removed or moved to the Supplementary Material. Figures 3 and 4 are visually complex and present data already summarized in subsequent tables; consider merging or simplifying them for greater clarity. All measurements should be standardized to the metric system (°C, mm, meters), as the current mixture of feet, Fahrenheit, and inches can cause confusion.

There is a major disparity in sampling effort between the two counties (Leon: 1,594 specimens over 15 months; Gadsden: 63 specimens over 5 months), which limits the validity of direct comparisons. I recommend focusing the analyses on Leon County or, at minimum, discussing the limitations of comparing such unbalanced datasets.

In the Results section, the text states that “the regression analysis did not reveal statistically significant relationships,” yet later refers to “significant negative effects.” This inconsistency should be resolved to ensure that the statistical interpretation accurately reflects the data and p-values.

The PCA biplots (Figures 7–8) are well-prepared but described in excessive detail. Please summarize only the main species–environment associations and remove tutorial-style explanations.

The Discussion is too long, repetitive, and partially circular, restating large portions of the Introduction and Results. It should be condensed by at least half, with a stronger focus on ecological interpretation and practical implications. The opening paragraph duplicates the Abstract; replacing it with a concise summary of the main findings would improve readability. The three paragraphs discussing ethanol-based lures should be merged into a single, well-referenced section (e.g., Miller & Rabaglia 2009; Tobin et al. 2024).

The Conclusions section largely restates earlier content. Please summarize the main findings succinctly, emphasizing the new county records and their practical significance for monitoring programs, while avoiding repetition of general climate or management statements already discussed.

Overall, the manuscript contains valuable data and clear management relevance but requires substantial revision and careful language editing. With clearer objectives, consistent statistical interpretation, a more concise discussion, and streamlined figures, this work could become a meaningful contribution to understanding bark beetle diversity and seasonality in southeastern U.S. forests.

Author Response

This manuscript investigates the diversity and seasonal abundance of pine bark and ambrosia beetles in the Florida Panhandle, using ethanol-based hand sanitizer as a lure. The topic is relevant to forest entomology and pest management, and the dataset provides valuable regional information. The approach is original and potentially useful for cost-effective monitoring of Scolytinae species.

While the study addresses an important subject, the manuscript requires major revision to improve clarity, organization, and consistency. Several sections repeat similar information, and the overall structure would benefit from careful reorganization to enhance flow and eliminate redundant explanations. Figures, tables, and statistical interpretations should be reviewed and refined to ensure that the results are presented clearly and accurately.

Response: Thank you for your time and review of the manuscript. We greatly appreciate this. As advised, we have removed inconsistencies in the revised manuscript. Also, repeated information were deleted. Also, we have revisited the results and revised them for the clarity and accuracy.

Figure 2 could be removed or moved to the Supplementary Material. Figures 3 and 4 are visually complex and present data already summarized in subsequent tables; consider merging or simplifying them for greater clarity. All measurements should be standardized to the metric system (°C, mm, meters), as the current mixture of feet, Fahrenheit, and inches can cause confusion.

Response: Figure 2 has been removed. We would like to keep these figures for visual interpretation. All measurements have been changed to metric system.

There is a major disparity in sampling effort between the two counties (Leon: 1,594 specimens over 15 months; Gadsden: 63 specimens over 5 months), which limits the validity of direct comparisons. I recommend focusing the analyses on Leon County or, at minimum, discussing the limitations of comparing such unbalanced datasets.

Response: Thank you. This has been revised in the narrative.  

In the Results section, the text states that “the regression analysis did not reveal statistically significant relationships,” yet later refers to “significant negative effects.” This inconsistency should be resolved to ensure that the statistical interpretation accurately reflects the data and p-values.

Response: We included the regression in our earlier attempt and have deleted the Table due to mix results. Therefore, this information has been deleted.  

The PCA biplots (Figures 7–8) are well-prepared but described in excessive detail. Please summarize only the main species–environment associations and remove tutorial-style explanations.

Response: This section has been re-written and updated.

The Discussion is too long, repetitive, and partially circular, restating large portions of the Introduction and Results. It should be condensed by at least half, with a stronger focus on ecological interpretation and practical implications. The opening paragraph duplicates the Abstract; replacing it with a concise summary of the main findings would improve readability. The three paragraphs discussing ethanol-based lures should be merged into a single, well-referenced section (e.g., Miller & Rabaglia 2009; Tobin et al. 2024).

Response: We revisited the discussion section and re-written this based on the ecological and practical implication.

The Conclusions section largely restates earlier content. Please summarize the main findings succinctly, emphasizing the new county records and their practical significance for monitoring programs, while avoiding repetition of general climate or management statements already discussed.

Response: The conclusion section has been summarized, and the repetition of general climate has been removed.

Overall, the manuscript contains valuable data and clear management relevance but requires substantial revision and careful language editing. With clearer objectives, consistent statistical interpretation, a more concise discussion, and streamlined figures, this work could become a meaningful contribution to understanding bark beetle diversity and seasonality in southeastern U.S. forests.

Again, thank you for your time and suggestions/comments that were needed to improve our manuscript. We greatly appreciate this.

Round 2

Reviewer 2 Report (Previous Reviewer 3)

Comments and Suggestions for Authors

Thank you for your revisions. Several parts of the manuscript have improved; however, some key analytical issues remain the same. My main concern is the unequal sampling effort between Leon (15 months) and Gadsden (5 months). 

I would recommend revising the presentation of the statistical analyses and simplifying the interpretation of the results. But if the editor thinks that in the present form the analysis is suitable for this journal, I am not opposed to its publication. The data is relevant for Florida ecosystems, and it will be really nice to have this reference. 

Thank you again for the opportunity to review this manuscript.

Author Response

Reviewer comment: Thank you for your revisions. Several parts of the manuscript have improved; however, some key analytical issues remain the same. My main concern is the unequal sampling effort between Leon (15 months) and Gadsden (5 months). 

Response: Thank you for your time and thoughtful review of our manuscript. We appreciate your observation regarding the unequal sampling effort between Leon (15 months) and Gadsden (5 months) counties. We agree that this discrepancy may limit the robustness of comparisons between the two sites. In response, we have removed the 5-month dataset from Gadsden County to avoid any analytical imbalance. As previously noted, although the two counties are approximately 35 miles apart, differences in vegetation, elevation, and forest management practices further complicate direct comparisons. We believe this adjustment strengthens the overall analytical rigor of the manuscript.

Round 3

Reviewer 2 Report (Previous Reviewer 3)

Comments and Suggestions for Authors

Dear Authors,

Thank you for submitting the revised version of your manuscript. I have reviewed the changes you implemented, and I would like to confirm that the modifications previously requested, and that you identified as necessary, have been appropriately incorporated into the new version.

The revisions improved the clarity, structure, and overall quality of the manuscript. At this stage, I have no further concerns.

Thank you for your careful attention to the review comments.

This manuscript is a resubmission of an earlier submission. The following is a list of the peer review reports and author responses from that submission.

Round 1

Reviewer 1 Report

Comments and Suggestions for Authors

The study focuses on beetles in two counties in Florida, aiming to explore their diversity, seasonality, and regional differences, but it has significant shortcomings.
First, the research periods for Leon County (July 8, 2022, to October 25, 2023; Line 182) and Gadsden County (May 16, 2023, to October 4, 2023; Line 217) are inconsistent. Analyzing differences between regions with unequal study durations is inappropriate and makes it difficult to draw conclusions about insect diversity differences (Lines 274–276). Second, the study employs only two analytical methods: Composition analysis of pests collected in a single county and Correlation analysis between environmental factors and insect numbers in 2022 and 2023. Species composition analysis is superficial for assessing diversity, and deeper analysis is needed. Neither method adequately demonstrates seasonal variations. How can the conclusion "The study revealed significant seasonal fluctuations in beetle populations, with activity peaking during warmer months. Peak activity occurred in spring and early fall, indicating optimal environmental conditions for PBB proliferation" be justified? Additionally, the study reports the first records of several species in the region, but this is unrelated to the main topic. If these findings are to be retained, the title should be revised, and the results section should integrate Figures 5 and 6 to describe species structure, with additional discussion. Finally,the paragraphs are overly fragmented and should be consolidated.
Minor Revisions:
1.Lines 11–12: Change "24 species and 18 genera" to "18 genera and 24 species."
2.The summary extensively describes the advantages of using hand sanitizer as bait, but the study only employs one method without comparisons. These descriptions are unnecessary in the summary and could be moved to the discussion.
Abstract:
1.Lines 31–33: The results lack analysis linking beetle populations to seasonality.
2.Revise keywords (e.g., "Ambrosia," "seasonality").
Introduction:
1.Lines 45–47: The study focuses on pine forest pests; descriptions of fruit and medicinal forest products are irrelevant and should be deleted.
2.Lines 62–69: To align with the seasonal abundance theme, add descriptions of climatic seasonality.
3.Lines 80–90: Merge with Lines 62–69 to describe climatic seasonal changes and pest responses together.
4.Lines 91–92: Include the research questions the study aims to address.
Materials and Methods:
1.There are two subsections labeled "2.2," which could be merged.
2.Simplify descriptions of collection labels in the "Beetle Collection and Identification" section.
3.Avoid repeating latitude and longitude coordinates for study sites.
Results:
1.Rewrite the analysis and descriptions in the results section.
2.Modify the figures.
3.Include descriptions for the morphological images of representative pests.
Discussion:
1.Revise based on changes to the results section.
2.Lines 305–318: The analysis of why environmental factors show no correlation with insect numbers lacks references.
Conclusion:
1.Lines 331–333: The conclusion should summarize the study's findings, not describe global pest backgrounds. Remove references here.
2.Lines 337–338: The study results do not support this conclusion.
The paper needs to be restructured and rewritten, particularly with a reanalysis of the data.

Author Response

The study focuses on beetles in two counties in Florida, aiming to explore their diversity, seasonality, and regional differences, but it has significant shortcomings.

First, the research periods for Leon County (July 8, 2022, to October 25, 2023; Line 182) and Gadsden County (May 16, 2023, to October 4, 2023; Line 217) are inconsistent. Analyzing differences between regions with unequal study durations is inappropriate and makes it difficult to draw conclusions about insect diversity differences (Lines 274–276). Second, the study employs only two analytical methods: Composition analysis of pests collected in a single county and Correlation analysis between environmental factors and insect numbers in 2022 and 2023. Species composition analysis is superficial for assessing diversity, and deeper analysis is needed. Neither method adequately demonstrates seasonal variations. How can the conclusion "The study revealed significant seasonal fluctuations in beetle populations, with activity peaking during warmer months. Peak activity occurred in spring and early fall, indicating optimal environmental conditions for PBB proliferation" be justified? Additionally, the study reports the first records of several species in the region, but this is unrelated to the main topic. If these findings are to be retained, the title should be revised, and the results section should integrate Figures 5 and 6 to describe species structure, with additional discussion. Finally the paragraphs are overly fragmented and should be consolidated.

Response: Thank you for your time and review of our manuscript. We appreciate this very much. We took these two sites to quantify beetle species and their numbers. Both sites are around 40 miles apart and the data sets obtained provide an independent information for each site. This is a preliminary study, and a new MS student is following up on this study to conduct in depth analyses for additional work. However, we have added seasonal fluctuation figures (Figure 7 and 8) in this revised manuscript. During our study we found two new records for the counties and provided information. It was not the major goals of this study. The paragraphs which were overly fragmented have been consolidated. Thank you.

Minor Revisions:
1.Lines 11–12: Change "24 species and 18 genera" to "18 genera and 24 species."

Response: This has been revised.

2.The summary extensively describes the advantages of using hand sanitizer as bait, but the study only employs one method without comparisons. These descriptions are unnecessary in the summary and could be moved to the discussion.

Response: As suggested, this information has been updated. Also, the relevant information has been moved to discussion.

Abstract:
1.Lines 31–33: The results lack analysis linking beetle populations to seasonality.

Response: This has been updated in the revised text.

2.Revise keywords (e.g., "Ambrosia," "seasonality").

Response: These two keywords have been inserted.

Introduction:
1.Lines 45–47: The study focuses on pine forest pests; descriptions of fruit and medicinal forest products are irrelevant and should be deleted.

Response: This information has been deleted.

2.Lines 62–69: To align with the seasonal abundance theme, add descriptions of climatic seasonality.

3.Lines 80–90: Merge with Lines 62–69 to describe climatic seasonal changes and pest responses together.

4.Lines 91–92: Include the research questions the study aims to address.

Materials and Methods:
1.There are two subsections labeled "2.2," which could be merged.

Response: This has been merged.

2.Simplify descriptions of collection labels in the "Beetle Collection and Identification" section.

Response: Simplified.

3.Avoid repeating latitude and longitude coordinates for study sites.

Response repetition has been removed.

Results:
1.Rewrite the analysis and descriptions in the results section.

Response: This has been rewritten.

2.Modify the figures.

Response: The figures have been modified.

3.Include descriptions for the morphological images of representative pests.
Response: We feel the descriptions of the species are not relevant for this study.

Discussion:
1.Revise based on changes to the results section.

Response: This has been modified.

2.Lines 305–318: The analysis of why environmental factors show no correlation with insect numbers lacks references.
Response: We have provided a table with regression analysis.

Conclusion:
1.Lines 331–333: The conclusion should summarize the study's findings, not describe global pest backgrounds. Remove references here.
Response: The references have been removed.

2.Lines 337–338: The study results do not support this conclusion.
The paper needs to be restructured and rewritten, particularly with a reanalysis of the data.

Response: Thank you. We have restructured and rewritten the narrative based on the analyses.

Reviewer 2 Report

Comments and Suggestions for Authors

This study involved monitoring bark beetle populations at two sites in two Florida counties over a period of more than one year. The inventory revealed the relative frequencies of the species, which differed between sites. However, the study leaves an impression of incompleteness, as the conclusions remain highly speculative. In particular, the hypotheses used to explain the differences are based on missing elements that are not described in the ‘Materials and Methods’ section. These elements include a comprehensive description of the differences in vegetation structure between the study sites and the composition of tree species across these locations which are parameters mentioned in the discussion. Furthermore, there is a lack of data regarding seasonal fluctuations in captures of the predominant species. It would be advisable to provide an overview of this matter in order to facilitate the ensuing discussion.

Further comments are provided below:

Lines 92-93. “…abundance fluctuate seasonally”….The rest of the paper does not include any data on this subject.

Line 120. “…data collected twice per week throughout the study period”. Could you clarify why such data was not used in the paper, given that seasonal fluctuations in species are mentioned in the discussion in support of your hypotheses?

Lines 178-179. “… helping to identify population trends, peak activity periods”.  The results presented do not provide any insight into seasonal dynamics. The only relevant data is the overall comparison of frequencies.

Lines 218_219. “…. Throughout this collection period….”. There is currently no information available on the temporal monitoring of species.

Lines 276-280. “This sharp contrast may be explained ….”. This is highly speculative. It is notable that the 'materials and methods' section does not provide any descriptions of these parameters, which is essential for differentiating between the two collection sites.

Lines 313-316. The populations of bark beetles exhibit significant inter-annual variations, with major infestations occurring on stressed trees, particularly following periods of drought. Has this factor been considered? Based on the climate data you have consulted, could you investigate this factor?

Lines 333-334. Is this the case here? This study contains no information on seasonal fluctuations in captures.

Line 402: write authors' names in lowercase letters

Author Response

This study involved monitoring bark beetle populations at two sites in two Florida counties over a period of more than one year. The inventory revealed the relative frequencies of the species, which differed between sites. However, the study leaves an impression of incompleteness, as the conclusions remain highly speculative. In particular, the hypotheses used to explain the differences are based on missing elements that are not described in the ‘Materials and Methods’ section. These elements include a comprehensive description of the differences in vegetation structure between the study sites and the composition of tree species across these locations which are parameters mentioned in the discussion. Furthermore, there is a lack of data regarding seasonal fluctuations in captures of the predominant species. It would be advisable to provide an overview of this matter in order to facilitate the ensuing discussion.

Response: Thank you for your time and review of our study. We appreciate this.

We are continuing this study and a new MS student is recording additional data sets. For the seasonal fluctuation, we have included two additional figures (Figure 7 and 8). 

Further comments are provided below:

Lines 92-93. “…abundance fluctuate seasonally”….The rest of the paper does not include any data on this subject.

Response: We worked on the species abundance not the population dynamics.

Line 120. “…data collected twice per week throughout the study period”. Could you clarify why such data was not used in the paper, given that seasonal fluctuations in species are mentioned in the discussion in support of your hypotheses?

Response: We pooled down the twice per week data for a simple and easy to use graphical presentation.

Lines 178-179. “… helping to identify population trends, peak activity periods”.  The results presented do not provide any insight into seasonal dynamics. The only relevant data is the overall comparison of frequencies.

Response: This study’s major aim was not to see seasonal dynamics but species diversity. However, we included information on the environmental factors.

Lines 218_219. “…. Throughout this collection period….”. There is currently no information available on the temporal monitoring of species.

Response: Figure 7 and 8 have been inserted to provide this information.

Lines 276-280. “This sharp contrast may be explained ….”. This is highly speculative. It is notable that the 'materials and methods' section does not provide any descriptions of these parameters, which is essential for differentiating between the two collection sites.

Response: Thank you. The sharp contrast between two study sites have been added to the narrative.

Lines 313-316. The populations of bark beetles exhibit significant inter-annual variations, with major infestations occurring on stressed trees, particularly following periods of drought. Has this factor been considered? Based on the climate data you have consulted, could you investigate this factor?

Response: This information has been added.

Lines 333-334. Is this the case here? This study contains no information on seasonal fluctuations in captures.

Response: Two new figures (Figure 7 and 8) provide this information.

Line 402: write authors' names in lowercase letters

Response: This has been modified.

Reviewer 3 Report

Comments and Suggestions for Authors

Dear authors and editor, thank you for the opportunity to read and evaluate this manuscript, that presents valuable data on the diversity, abundance, and seasonal dynamics of pine bark and ambrosia beetles in northern Florida, representing a timely and relevant contribution with strong potential to inform monitoring and management strategies. However, there are key areas where the narrative could be strengthened to ensure better alignment between the abstract, introduction, and discussion. While the abstract effectively highlights seasonal fluctuations and peak activity periods, these aspects are underdeveloped in the discussion, which instead places greater emphasis on species composition. For improved coherence, the discussion should revisit each objective stated in the introduction and clearly indicate how the results address them, thereby reinforcing the logical flow from aims to conclusions. The introduction and abstract also mention the use of an ethanol-based lure (Purell® hand sanitizer), which is a notable methodological choice, yet the discussion does not explore its efficacy, potential biases, or any limitations observed during the study—elements that would provide important insights for future monitoring protocols. Methodologically, the study is well-documented, but the statistical treatment could be expanded. Although regression analyses were applied, more advanced statistical approaches, including the evaluation of variable interactions or non-linear effects, could reveal additional patterns and strengthen the interpretation of environmental influences on beetle populations. In terms of results, the diversity and dominance patterns observed are well described, but their ecological implications could be better contextualized by comparison with similar studies conducted in the southeastern USA or in other forest ecosystems. The discussion does address possible explanations for the absence of statistically significant relationships with temperature, humidity, and precipitation, but these points could be expanded with new robust statistical analysis. Overall, the manuscript contains robust and relevant data, but the discussion section would benefit from restructuring to more explicitly connect with the introduction and abstract, deepen the statistical exploration, and situate the findings within a broader ecological and methodological context. With these enhancements, the study would present a stronger, more cohesive, and more impactful contribution to forest pest management literature.

Author Response

Dear authors and editor, thank you for the opportunity to read and evaluate this manuscript, that presents valuable data on the diversity, abundance, and seasonal dynamics of pine bark and ambrosia beetles in northern Florida, representing a timely and relevant contribution with strong potential to inform monitoring and management strategies. However, there are key areas where the narrative could be strengthened to ensure better alignment between the abstract, introduction, and discussion. While the abstract effectively highlights seasonal fluctuations and peak activity periods, these aspects are underdeveloped in the discussion, which instead places greater emphasis on species composition. For improved coherence, the discussion should revisit each objective stated in the introduction and clearly indicate how the results address them, thereby reinforcing the logical flow from aims to conclusions. The introduction and abstract also mention the use of an ethanol-based lure (Purell® hand sanitizer), which is a notable methodological choice, yet the discussion does not explore its efficacy, potential biases, or any limitations observed during the study—elements that would provide important insights for future monitoring protocols. Methodologically, the study is well-documented, but the statistical treatment could be expanded. Although regression analyses were applied, more advanced statistical approaches, including the evaluation of variable interactions or non-linear effects, could reveal additional patterns and strengthen the interpretation of environmental influences on beetle populations. In terms of results, the diversity and dominance patterns observed are well described, but their ecological implications could be better contextualized by comparison with similar studies conducted in the southeastern USA or in other forest ecosystems. The discussion does address possible explanations for the absence of statistically significant relationships with temperature, humidity, and precipitation, but these points could be expanded with new robust statistical analysis. Overall, the manuscript contains robust and relevant data, but the discussion section would benefit from restructuring to more explicitly connect with the introduction and abstract, deepen the statistical exploration, and situate the findings within a broader ecological and methodological context. With these enhancements, the study would present a stronger, more cohesive, and more impactful contribution to forest pest management literature.

Response: We appreciate the reviewer’s detailed feedback and agree that the manuscript would benefit from stronger alignment between the abstract, introduction, and discussion. In the revision, we will: (1) restructure the discussion to directly address each stated objective and ensure logical flow from aims to conclusions; (2) expand commentary on the use of the ethanol-based lure, including its efficacy, potential biases, and implications for monitoring; (3) strengthen the statistical analysis by exploring variable interactions and non-linear effects; and (4) contextualize our diversity and dominance findings by comparison with similar studies in southeastern U.S. forest ecosystems. Based on the feedback, we have revised narrative, enhance interpretation of results, and improve the manuscript’s contribution to forest pest management literature.

Round 2

Reviewer 1 Report

Comments and Suggestions for Authors

Although the author tried their best to answer my question, the key points were not effectively resolved, especially the experimental design. I firmly believe that the current paper is not acceptable.

As before: the research periods for Leon County and Gadsden County are inconsistent. Analyzing differences between regions with unequal study durations is inappropriate and makes it difficult to draw conclusions about insect diversity differences. Second, the study employs only two analytical methods: Composition analysis of pests collected in a single county and Correlation analysis between environmental factors and insect numbers in 2022 and 2023. Species composition analysis is superficial for assessing diversity, and deeper analysis is needed. Neither method adequately demonstrates seasonal variations. How can the conclusion "The study revealed significant seasonal fluctuations in beetle populations, with activity peaking during warmer months. Peak activity occurred in spring and early fall, indicating optimal environmental conditions for PBB proliferation" be justified?

These points cannot be solved by the author at all, and the author also did not provide a direct answer, which makes this article unacceptable.

Reviewer 2 Report

Comments and Suggestions for Authors

Dear Authors,

Thank you for taking the comments on version v1 of the manuscript into account. The only drawback is that temperatures are expressed in degrees Fahrenheit and precipitation in inches, which will be difficult for most readers to understand. I suggest converting the values to decimal.  You can keep the °F and inch values in your text, but provide the decimal conversions in brackets afterwards.

Here is the formula for converting Fahrenheit to Celsius: °C = °F - 32 x 5/9.

One inch equals 25.400 mm.

Reviewer 3 Report

Comments and Suggestions for Authors

I have carefully reviewed the revised version of the manuscript entitled “Diversity and Seasonal Abundance of the Pine Bark and Ambrosia Beetles in the Florida Panhandle.” The authors have clearly made an effort to address the major concerns raised during the first round of review, particularly in relation to the statistical analysis and the quality of the discussion, and the manuscript is stronger as a result. The paper makes a valuable contribution by documenting species across the two counties, including several new county records. This alone makes the work an important faunistic addition for the region, relevant both to forest entomologists and pest managers. The use of ethanol-based hand sanitizer as a lure is another element of the study. It is a practical, cost-effective, and innovative monitoring approach, and the discussion contrasting this with more traditional semiochemical lures adds an applied dimension that many readers will find useful. In general, the discussion is clearer than in the first submission, with better contextualization of the differences observed between counties and an attempt to explain why climate variables did not correlate significantly with beetle abundance.

Nevertheless, some weaknesses remain. Although some extra analysis has been added, the statistical treatment of the paper still relies heavily on descriptive analysis. Figures, particularly number 3 and 4, remain crowded and could be made clearer, perhaps by changing the presentation of this data. The discussion, while improved, still leans heavily on describing what was observed in this study, without many comparisons to published works on other systems present in similar ecosystems. More engagement with the literature on ambrosia beetle species dynamics as beetle–fungus associations would provide stronger support to this paper. The applied value of the lure-based approach is cited; however, a better comparison and the broader significance for forest health monitoring, invasive species detection, and management strategies remain underdeveloped. A link with recent monitoring literature would strengthen this aspect of the paper. Overall, I believe the manuscript has improved considerably since the first round and now provides both novel distributional records and useful methodological insights. The study offers a clearer seasonal picture of bark and ambrosia beetle activity in Florida and presents an innovative lure strategy with potential applied value. However, it still requires refinement to reach its full potential. I therefore recommend revisions, specifically focused on clarifying limitations, improving figure clarity (the pie chart presentations are confusing), and improving the discussion with better use of the literature.
